# BiFC Method Based on Intraorganellar Protein Crowding Detects Oleate-Dependent Peroxisomal Targeting of *Pichia pastoris* Malate Dehydrogenase

**DOI:** 10.3390/ijms22094890

**Published:** 2021-05-05

**Authors:** Jean-Claude Farré, Paul Li, Suresh Subramani

**Affiliations:** Section of Molecular Biology, Division of Biological Sciences, University of California, San Diego, CA 92093-0322, USA; pyl001@ucsd.edu

**Keywords:** redox balance, NADH shuttle, peroxisomal malate dehydrogenase, environment-dependent peroxisomal targeting, intraorganellar protein crowding

## Abstract

The maintenance of intracellular NAD^+^/NADH homeostasis across multiple, subcellular compartments requires the presence of NADH-shuttling proteins, which circumvent the lack of permeability of organelle membranes to these cofactors. Very little is known regarding these proteins in the methylotrophic yeast, *Pichia pastoris*. During the study of the subcellular locations of these shuttling proteins, which often have dual subcellular locations, it became necessary to develop new ways to detect the weak peroxisomal locations of some of these proteins. We have developed a novel variation of the traditional Bimolecular Fluorescence Complementation (BiFC), called divergent BiFC, to detect intraorganellar colocalization of two noninteracting proteins based on their proximity-based protein crowding within a small subcellular compartment, rather than on the traditional protein–protein interactions expected for BiFC. This method is used to demonstrate the partially peroxisomal location of one such *P. pastoris* NADH-shuttling protein, malate dehydrogenase B, only when cells are grown in oleate, but not when grown in methanol or glucose. We discuss the mode of NADH shuttling in *P. pastoris* and the physiological basis of the medium-dependent compartmentalization of *Pp*MdhB.

## 1. Introduction

The Bimolecular Fluorescence Complementation (BiFC) assay is a protein proximity-based method developed initially to directly visualize protein–protein interaction in vivo using live-cell imaging or fixed cells. BiFC was first developed using two nonfluorescent protein fragments of the enhanced yellow fluorescent protein (EYFP), which is an *Aequorea victoria* GFP variant protein [1]. The YFP was later improved by introducing mutations to reduce its sensitivity to pH and chloride ions, thus generating a super-enhanced YFP, named Venus fluorescent protein, without reduction in fluorescence at 37 °C, as often observed with EYFP [2]. The fluorescence signal is regenerated by complementing two nonfluorescent protein fragments (e.g., the Venus N-terminal 1–158 amino acids, called VN, and its C-terminal 159–239 amino acids, named VC) that are brought together by the interaction between their respective fusion partners.

Here we use BiFC somewhat differently, in a method dubbed “divergent BiFC”, to detect colocalization of two noninteracting proteins fused to VN and VC, respectively, within a restricted subcellular compartment, the peroxisome. We show that fluorescence can be generated by proximity-based protein crowding in peroxisomes.

We used the divergent BiFC method to understand the NADH-shuttling proteins that are normally localized to three subcellular compartments, namely, the mitochondria, peroxisomes, and the cytosol [3]. Mechanisms for the maintenance of the NAD^+^/NADH redox balance are conserved in evolution, and the sum of the NAD^+^ and NADH concentration is around 1 mM in *Saccharomyces* [4]. This homeostasis requires that as NAD^+^ is reduced, NADH must be oxidized intracellularly. The pathways by which this homeostasis is achieved have been reviewed elsewhere [4]. What is important to note, however, is that NADH metabolism occurs in different subcellular compartments, whose membranes are not permeable to these cofactors. To circumvent this challenge, NADH-shuttling proteins exist in eukaryotic cells that facilitate the efficient transfer between subcellular compartments, such as the mitochondria, where NADH is used to generate ATP, peroxisomes where NADH is generated in yeast cells grown on fatty acids as the sole carbon source and the cytosol [4,5]. In the methylotrophic yeast, *P. pastoris*, none of the shuttling mechanisms that transport NADH have been described, and the proteins responsible have not been identified.

We demonstrate the dual localization to mitochondria and the cytosol for two such *Pichia pastoris* NADH-shuttling proteins, glycerol-3-phosphate dehydrogenase (*Pp*GpdA) and one isoform of malate dehydrogenase A (*Pp*MdhA) to both mitochondria and cytosol. Additionally, using the divergent BiFC method, we detected the peroxisomal and cytosolic localization of another NADH-shuttling protein, malate dehydrogenase B (*Pp*MdhB). We discuss the physiological role of the peroxisomal localization of this particular malate dehydrogenase isoform (*Pp*MdhB) only in oleate-, but not in methanol- or glucose-grown cells.

## 2. Results

### 2.1. NADH-Shuttling Proteins in P. pastoris

*S. cerevisiae* encodes three malate dehydrogenases (Mdh: Mdh1, Mdh2, and Mdh3) that display strong sequence similarities between them, the most significant of which are the presence of either a mitochondrial or a peroxisomal targeting signal (PTS) (Appendix A). Similarly, the two glycerol-3-phosphate dehydrogenases (Gpd: Gpd1 and Gpd2) share a strong protein sequence homology, but one of them has a PTS (Appendix A).

To identify the *P. pastoris* counterpart of these dehydrogenases, we searched the *P. pastoris* GS115 database performing a protein–protein BLAST (BLASTP) using the three isoforms of Mdh and two isoforms of Gpd from *S. cerevisiae* as a query. We found that *P. pastoris* has two isoforms of Mdh and only one isoform of Gpd (Appendix A). Since we did not know which isoforms in *P. pastoris* correspond to those in *S. cerevisiae*, we refer to them as *Pp*MdhA, *Pp*MdhB, and *Pp*GpdA.

We analyzed the *P. pastoris* NADH-shuttling proteins sequences manually and using the prediction algorithm of the PTS1 predictor version 1 (http://mendel.imp.ac.at/pts1/ accessed on 14 March 2021), but no obvious SKL tripeptide, or its variant, was found at their C-termini (Appendix A and Table 1). A PTS score equal to or greater than 0 would potentially have a predicted PTS1, while a PTS score under –10 would not be predicted to be a PTS1 sequence [6]. Based on this scoring system, all of the Mdh and Gpd proteins listed for *P. pastoris* and *S. cerevisiae* scored below –10 and did not have a predicted PTS1, except for *Sc*Mdh3 that has a PTS score above 0, as well as an SKL as the PTS1 at its C-terminus (Table 1).

When analyzing the N-terminus of the NADH-shuttling protein homologs, no sequences corresponding to the PTS2 consensus [7], (R/K)-(L/V/I)-(X)_5_-(H/Q)-(L/A) or (R/K)-(L/V/I/Q)-X-X-(L/V/I/H/Q)-(L/S/G/A/K)-X-(H/Q)-(L/A/F), were found in the *P. pastoris* Mdh or Gpd proteins using EMBOSS fuzzpro that can search for the different combinations of the consensus sequences [8]. Only *Sc*Gpd1 has a PTS2 at the N-terminus with the consensus sequence (RLNLTSGHL) (Appendix A). The closest PTS2 consensus near the N-terminus of the *P. pastoris* proteins had a match of only 6 out of 9 amino acids for *Pp*GpdA, *Pp*MdhA, and *Pp*MdhB.

However, *Pp*MdhA and *Pp*GpdA have a high probability (above the threshold score of 0.5) of being targeted to the mitochondria and contain an identifiable mitochondrial presequence at the N-terminus based on the MitoProtII program (https://bio.tools/MITOPROT_II accessed on 14 March 2021) [9]. More specifically, *Pp*MdhA has a probability of 0.948 and an 18 amino acid, mitochondrial presequence (MLSTIAKRQFSSSASTA), while *Pp*GpdA has a probability of 0.6471 for being translocated to the mitochondria with a 24 amino acid, mitochondrial presequence (MYLTSTVRALPVHFFRSRHCIRT) (Table 1). *Sc*Mdh1 and *Sc*Gpd2 have a probability greater than 0.9 and are known to have a mitochondrial localization, which is similar to *Pp*MdhA that also has a high probability for mitochondrial targeting. *Pp*GpdA has a lower probability than the *S. cerevisiae* mitochondrial NADH-shuttling proteins (Mdh1 and Gpd2), but a higher probability than cytosolic or peroxisomal shuttling dehydrogenases (*Sc*Mdh2, *Sc*Mdh3, and *Sc*Gpd1). In conclusion, *Pp*MdhA has a high chance of being a mitochondrial enzyme and could be the homolog of *Sc*Mdh1. *Pp*MdhB does not have a distinct targeting signal, so the putative localization cannot be elucidated in silico.

### 2.2. Localization of PpMdhA-GFP, PpMdhB-GFP, and PpGpdA-GFP in Different Carbon Sources

We found *Pp*MdhA and *Pp*GpdA, both lacking an identifiable PTS1 and PTS2 sequence, to have a high likelihood of mitochondrial targeting in silico. Of the three dehydrogenases, *Pp*MdhB has the lowest probability for mitochondrial targeting, hence the best candidate to be at least partially localized to peroxisomes. In *S. cerevisiae*, Mdh3 is important for intraperoxisomal redox balance of NAD(H) in oleate, but not in other media. If *P. pastoris* shares a similar shuttling mechanism with *S. cerevisiae* for transporting NADH from the peroxisome to the mitochondria and maintaining the intraperoxisomal redox balance in oleate, then it would be important to study the localization of *Pp*MdhA, *Pp*MdhB, and *Pp*GpdA in glucose, methanol, and oleate.

To localize the NADH-shuttling proteins in *P. pastoris*, we fused GFP to the C-terminus of *Pp*MdhA/B and *Pp*GpdA, and expressed these GFP fusion proteins from their endogenous promoters in wild-type cells (Figure 1). We also transformed a mitochondrial and peroxisomal marker, Tom20-mRFP and BFP-SKL, respectively, into the wild-type strains containing the shuttling dehydrogenase-GFP fusion proteins to determine if there was colocalization at the mitochondria or peroxisomes. Both *Pp*MdhA-GFP and *Pp*GpdA-GFP colocalized with the mitochondrial marker in glucose, methanol, and oleate media, but also appeared to have partially cytosolic fluorescence, suggesting that these two proteins may possess dual localization in *P. pastoris* in all three media (Figure 1A,B).

In contrast, *Pp*MdhB-GFP differed in that it appeared cytosolic in all three media with no obvious intraorganelle localization (Figure 1C). Based on the cytosolic and mitochondrial localization, supported by its likelihood for mitochondrial targeting, *Pp*GpdA is likely the homolog of *Sc*Gpd2. The mitochondrial localization confirmed the high probability for mitochondrial targeting for *Pp*MdhA, making it likely to be the homolog of *Sc*Mdh1. Finally, based on the cytosolic localization, *Pp*MdhB appeared to be like *Sc*Mdh2.

Based on the localization of NADH-shuttling proteins in all the media tested in *P. pastoris*, it would appear that NADH might not shuttle out of the peroxisome, as neither *Pp*Mdh isoforms, nor *Pp*Gpd, had any detectable peroxisomal localization. However, given the importance and conservation of the NADH shuttle and peroxisomal NADH production, we sought a more sensitive technique to detect small pools of peroxisomally-localized NADH-shuttling proteins.

### 2.3. An Assay to Improve Detection of Peroxisomally-Localized Proteins

We hypothesized that a small fraction of the NADH-shuttling proteins might localize at peroxisomes, but that this localization was probably drowned out by the predominantly cytosolic fraction. To detect a putative small peroxisomal fraction, we used the bimolecular fluorescence complementation (BiFC) technique.

BiFC uses a fluorescent Venus protein (V) that is split into two, nonfluorescent halves, called VN and VC, which are typically fused to two proteins that may physically interact or be in close proximity to bring VN and VC together to reconstitute GFP, and yield fluorescence [10]. For investigating the localization of the NADH-shuttling proteins, we took advantage of the latter case. We targeted the VN moiety to the peroxisome through the most efficient PTS1 pathway [11], by fusing the ePTS1 (LGRGRRSKL) to the C-terminus of VN and fusing the NADH-shuttling proteins with VC. To increase the chances that VN and VC find each other and yield fluorescence, we overexpressed both fusion proteins. The PTS1 pathway is extremely efficient, and a large fraction of VN-ePTS1 was expected to localize to peroxisomes [11]. If a small fraction of the NADH-shuttling proteins fused to VC is also imported to the peroxisomes, the fluorescence at the peroxisomes should exceed that in the cytosol, allowing detection by regular fluorescence microscopy (Figure 2).

To prove the concept of the divergent BiFC assay, we coexpressed together with VN-ePTS1, a peroxisomal noninteracting protein, thiolase (Pot1), fused with VC at its C-terminus. Thiolase is synthesized in the cytosol and imported via the PTS2 pathway into peroxisomes, where it catalyzes the last step of the β-oxidation pathway [12]. Since the PTS2 nonapeptide is near the amino terminus of Pot1, the VC moiety was fused at the C-terminus. Thus, both the Pot1-VC with its PTS2, and VN-ePTS1 with its PTS1 can still be recognized by their respective PTS receptors, Pex7 and Pex5, respectively, and imported into peroxisomes.

The overexpression and peroxisomal crowding of these noninteracting proteins was expected to bring together VN and VC to yield fluorescence. The plasmid coexpressing Pot1-VC and VN-ePTS1 was integrated into the genome of *P. pastoris* wild-type cells expressing peroxisomal Pex3-mRFP from its own promoter and analyzed by fluorescence microscopy after induction in glucose, methanol, or oleate media (Figure 3). *P. pastoris* glucose-grown cells contain a single small peroxisome (area: ~0.05 μm^2^), and when cells are grown in media requiring peroxisome metabolism, the peroxisome size and number increase [13]. The most commonly used media for peroxisome proliferation in *P. pastoris* are oleate and methanol. When cells are grown in oleate media overnight, peroxisome number and area increase to ~6 and to ~0.08 μm^2^, respectively, and peroxisomes are dispersed in the cytosol near the cell periphery. Similarly, when cells are grown overnight in methanol media, peroxisome number and area increase to ~4 and to ~0.4 μm^2^, respectively, and are clustered.

As hypothesized, the overexpression of the two peroxisomal chimera proteins Pot1-VC and VN-ePTS1 in wild-type cells, showed a strong fluorescence at the peroxisome in every medium, confirmed by the colocalization with Pex3-mRFP, indicating the two nonfluorescent halves of the Venus proteins were reconstituted in the peroxisomes, most likely due to their proximity created by their high abundance in a confined area (Figure 3A–C).

To confirm that the peroxisomal fluorescence observed in our divergent BiFC assay was a consequence of the Venus reconstitution in the peroxisomal matrix and not a weak fluorescence of the Venus moiety itself, we knocked out different essential components of peroxisomal matrix protein import. With this purpose, we transformed the plasmid coexpressing Pot1-VC and VN-ePTS1, along with Pex3-mRFP to mark peroxisomes, into cells mutated for the PTS2 receptor (Δ*pex7*) and the peroxisomal receptor docking protein (Δ*pex14*) and tested them in glucose, methanol, and oleate (Figure 3).

When we analyzed cells lacking the PTS2 receptor in the three media, no fluorescence was observed at the peroxisome, although we noticed some cytosolic fluorescence (Figure 3A–C). These results were expected, at least for the lack of peroxisomal fluorescence, as Pot1 was not expected to be imported into the peroxisome without its PTS2 receptor, Pex7. However, the cytosolic fluorescence was unexpected because VN-ePTS1 should be imported into the peroxisome in the presence of the PTS1 receptor, Pex5, which might suggest that a fair amount of the VN-ePTS1 synthesized in the cytosol remained there.

When we analyzed cells lacking the peroxisomal receptor docking protein Pex14, no peroxisomal fluorescence was observed in any media as expected (Figure 3A–C), because, in the absence of Pex14, neither Pex5 nor Pex7 would be expected to dock at the peroxisomal membrane to import their respective cargoes. Similar to the result in the Δ*pex7* cells, a cytosolic fluorescence was observed in the Δ*pex14* cells, although in this case, the result was highly expected as both VN-ePTS1 and Pot1-VC were not supposed to be imported. In addition, the fluorescence in the cytosol indicated the constitutive *HTX1* promoter is strong enough to crowd even the cytosol with the two Venus moieties.

To verify the specificity of the divergent BiFC assay, we coexpressed GAPDH-VC and VN-ePTS1 in wild-type cells and observed that in all three media, the fluorescence was localized in the cytosol, and no colocalization with Pex3-mRFP was observed (Figure 3D). To verify that Gapdh-VC was capable of being transported to, and interacting with VN-ePTS1 in, the peroxisomes, we appended a PTS1 at its C-terminus (Gapdh-VC-SKL) and observed a clear peroxisomal signal (Appendix A). This control proves that the lack of peroxisomal fluorescence of Gapdh-VC is because this protein does not piggyback into peroxisomes with VN-ePTS1.

Thus, the divergent BiFC assay worked as expected with a strong peroxisomal fluorescence for Pot1, which could be abolished by the deletion of *PEX7* and *PEX14* genes, and no peroxisomal fluorescence was observed when we swapped Pot1 with the cytosolic protein Gapdh in wild-type cells.

### 2.4. Testing the Localization of the NADH-Shuttling Proteins Using the BiFC Assay

We then tested the localization of our NADH-shuttling proteins by transforming *Pp*GpdA-, *Pp*MdhA-, or *Pp*MdhB-VC and VN-ePTS1 plasmids, along with those expressing peroxisomal markers into wild-type cells. These fusions with VC at the C-terminus of the NADH-shuttling proteins would test whether these proteins had either an N-terminal or internal PTS, while also potentially blocking peroxisomal import by a C-terminal PTS.

In wild-type cells, we observed mitochondrial fluorescence for *Pp*GpdA and *Pp*MdhA in glucose, methanol, and oleate (Figure 4A,B). Peroxisomal fluorescence was not observed, suggesting no colocalization with Pex3-mRFP, as expected if VN-ePTS1 was imported into the peroxisome and *Pp*GpdA or *Pp*MdhA was imported to the mitochondria, as predicted by our sequence analysis.

However, we did not expect detectable mitochondrial fluorescence, unless overexpressed VN-ePTS1 was somehow mislocalized (partially) to mitochondria. A possible explanation would be that due to the overexpression, sufficient amounts of the *Pp*Gpd-VC or *Pp*MdhA-VC fusions come in contact with VN-ePTS1 in the cytosol, yielding fluorescence, and due to the irreversible nature of the BiFC interaction get imported to, or accumulated at, the mitochondrial outer membrane.

More importantly, in wild-type cells, *Pp*MdhB displayed cytosolic fluorescence in glucose and methanol, but in oleate, it was partially peroxisomal and cytosolic (Figure 5A). Similar to Gapdh, *Pp*MdhB exhibited cytosolic fluorescence, as *Pp*MdhB remained mostly in the cytosol and reconstituted the Venus protein with a portion of VN-ePTS1 synthesized in the cytosol, indicating that *Pp*MdhB is a cytosolic protein. In oleate, *Pp*MdhB displayed cytosolic and peroxisomal fluorescence colocalized with Pex3-mRFP, suggesting that *Pp*MdhB has a dual localization in the cytosol and peroxisomes in this medium.

In order to further study the peroxisomal import of *Pp*MdhB, we transformed *Pp*MdhB-VC and VN-ePTS1, as well as Pex3-mRFP, into Δ*pex7* and Δ*pex14* cells. In oleate, similar to Pot1 in Δ*pex7* and Δ*pex14* cells, *Pp*MdhB displayed cytosolic fluorescence in Δ*pex7* cells, suggesting its peroxisomal import in a manner that is dependent on the PTS2 pathway (Figure 5B). As expected, no peroxisomal fluorescence was observed for *Pp*MdhB when import to this organelle was blocked using Δ*pex14* cells (Figure 5B).

### 2.5. VC Fused to the N-Terminus of PpMdhB Is Not Imported to Peroxisomes

To analyze whether *Pp*MdhB may have a noncanonical PTS1 created by an unknown mechanism, such as ribosomal readthrough or mRNA splicing, we also fused VC to the N-terminus of *Pp*MdhB (VC-*Pp*MdhB) and transformed VC-*Pp*MdhB and VN-ePTS1, as well as Pex3-mRFP, into wild-type cells. We found VC-*Pp*MdhB to be cytosolic in glucose, methanol, and oleate (Appendix A). The lack of peroxisomal localization of VC-*Pp*MdhB suggests that no PTS1 was exposed, but also indicates that an N-terminal tag in *Pp*MdhB hampers its interaction with the PTS receptor or carrier in wild-type cells.

### 2.6. Independent Confirmation of the Peroxisomal Localization of PpMdhB-GFP in Oleate

Because the peroxisomal fraction of PpMdhB may have eluded detection when it was expressed from its own promoter, we expressed *Pp*MdhB-GFP from the alcohol oxidase promoter (P_AOX_) that is tightly regulated, in an effort to reduce the cytosolic fluorescence and to make the peroxisomal fluorescence more apparent. P_AOX_ is induced in the presence of methanol, strongly repressed by glucose, and not expressed in oleate. We transformed the plasmid containing *Pp*MdhB-GFP expressed from P_AOX_ into wild-type cells, as well as plasmids expressing Tom20-mRFP and BFP-SKL as mitochondrial and peroxisomal markers, respectively, for colocalization studies.

As a control, we induced expression of *Pp*MdhB-GFP with methanol and found it to be fully cytosolic in cells grown overnight in methanol (Figure 6A). There was no colocalization with either peroxisomal or mitochondrial markers as expected. To control the levels of *Pp*MdhB, we turned on the expression of *Pp*MdhB and peroxisome proliferation for only 2 h in methanol medium, turned off *Pp*MdhB expression using glucose in the medium for 1 h, and finally transferred the wild-type cells to oleate medium overnight for peroxisome proliferation (Figure 6B). Glucose should repress the *AOX* promoter and turn off peroxisome proliferation in methanol that usually results in clustered peroxisomes as observed in our control. With a limited production of *Pp*MdhB, we expected a portion of *Pp*MdhB would remain in the cytosol, while another portion would localize to peroxisomes in oleate. Using this approach, we were indeed able to detect the peroxisomal isoform of *Pp*MdhB colocalizing with BFP-SKL (Figure 6B).

These results show convincingly that the peroxisomal fluorescence of *Pp*MdhB, observed using the divergent BiFC assay, could be independently reproduced by controlling *Pp*MdhB-GFP expression levels resulting in peroxisomal *Pp*MdhB in oleate medium.

## 3. Discussion

While most proteins localized to the peroxisomal matrix have defined PTS sequences responsible for their localization to the peroxisome matrix, and the subcellular localization of such proteins is often conserved through evolution, determining the peroxisomal localization and its mechanisms has been challenging for a subset of peroxisomal matrix proteins. These may be divided into five broad categories—(1) those that have no obvious PTS, but might be transported into peroxisomes in a “piggyback” fashion in association with another protein with a bona fide PTS [14]; (2) proteins whose PTS is generated by unusual mechanisms, such as ribosomal readthrough past a canonical stop codon [15,16], alternative splicing [17,18,19] or the use of alternative transcription start sites [20] or translational start sites [21]; (3) proteins with dual localization, including peroxisomes [8], where the detection of a small peroxisomal pool may require techniques with high sensitivity [22]; (4) dynamic, environment-specific peroxisomal localization that might be missed by experimental analysis of the protein under the wrong environmental conditions [14,23,24,25]; and (5) post-translational activation or inhibition of a PTS [23,26].

The maintenance of the cellular redox balance between NAD^+^, and its reduced form NADH, is critical for many metabolic processes, as exemplified by the fact that the oxidation and reduction of NAD^+^ and NADH drive both catabolic and anabolic reactions [4]. Cellular NAD^+^ levels are maintained by both de novo and salvage biosynthetic pathways [27]. The majority of NADH formation is connected to biomass production: Synthesis of proteins, nucleic acids, and highly reduced lipids [4]. In yeasts, since all fatty acid oxidation occurs in peroxisomes, this organelle is an important source of NADH when cells are grown in carbon sources, such as oleate. The peroxisomal β-oxidation of fatty acids consumes NAD^+^, yielding NADH, which needs to shuttle out of the peroxisome to maintain a peroxisomal redox balance. Once NADH is in the cytosol, it shuttles to the mitochondria and is subsequently transferred to complex I in the electron transport chain, contributing to the production of ATP. NADH is also produced, albeit not intraperoxisomally, by methyltrophic yeasts grown in methanol, whose oxidation occurs by a process that reduces NAD^+^, yielding 2 NADH in the cytosol, which shuttle to mitochondria for ATP production as well. Thus, we expected that some of the NADH-shuttling proteins in *P. pastoris* would be peroxisomally localized at least during growth in oleate, which produces intraperoxisomal NADH, but we were surprised to find that this pool was not detectable by traditional subcellular localization methods.

We, therefore, developed a sensitive method to detect small pools of peroxisomally-localized proteins. The method is based on a divergent use of the BiFC method, normally used to study interactions between proteins. It is based on the concept of protein crowding of two noninteracting proteins fused to N- and C-terminal halves of the fluorescent Venus protein, rather than their physical interaction, within a small subcellular compartment to generate a fluorescence signal. Such proximity, reflected by the presence of fluorescence, would then indicate that the two proteins must be colocalized. The divergent BiFC method used here directly visualizes the spatial and temporal interaction between two colocalized, noninteracting proteins in vivo. The fluorescence signal provides a sensitive readout for detecting protein colocalization even at a low expression level comparable to that of the endogenous proteins and detects peroxisomal localization even in the background of some cytosolic labeling.

After demonstrating a proof-of-concept with peroxisomally-localized Pot1-VC and VN-ePTS1 proteins, we used this divergent BiFC method to determine whether any of the NADH-shuttling proteins in *P. pastoris* were localized in peroxisomes. Our finding was that only *Pp*MdhB, but not *Pp*MdhA or *Pp*Gpd1, was partially peroxisomal and also cytosolic, but only in cells grown in oleate, and not in methanol or glucose. The localization of a protein to peroxisomes only in oleate is not unprecedented because, in *S. cerevisiae*, aspartate aminotransferase (Aat2) is cytosolic in glucose-grown cells, but peroxisomal in cells grown in oleate [28]. Indeed, recent studies show that several yeast proteins, particularly those involved in the β-oxidation of fatty acids, exhibit a priority for peroxisomal targeting in oleate, in comparison to glucose [29].

*Pp*MdhA-GFP and *Pp*GpdA-GFP appeared to not only have partially cytosolic fluorescence, but also colocalized with the mitochondrial marker in glucose, methanol, and oleate media, consistent with the presence of high scoring mitochondrial targeting signals in these proteins. The organellar location of all three proteins was also confirmed independently using GFP fusions of these proteins, and particularly for *Pp*MdhB expressed in the pulse-chase experiment from the inducible *AOX* promoter in methanol, and that was then turned off in oleate, thereby validating the divergent BiFC assay, as well as our conclusions.

### 3.1. Physiological Roles of NADH-Shuttling Proteins in P. pastoris

Our work suggests that *Pp*GpdA may be the homolog of *Sc*Gpd2 based on its mitochondrial localization and may play a role in shuttling NADH from the cytosol to the mitochondria. In contrast to *S. cerevisiae*, *P. pastoris* does not appear to be shuttling NADH out of the peroxisome through the glycerol 3-phosphate shuttle, as peroxisomal fluorescence and colocalization with peroxisomal markers for *Pp*GpdA were not observed. *Pp*MdhA may be the homolog of *Sc*Mdh1 based on its mitochondrial localization and may play a role in the TCA cycle in the mitochondria, in addition to regenerating NADH in the mitochondria that were shuttled in from the cytosol through the malate shuttle. Overall, it appears that NADH may be shuttled from the cytosol to the mitochondria through the malate dehydrogenase shuttle and/or the glycerol-3 phosphate dehydrogenase shuttle in *P. pastoris* (Figure 7).

In addition to the two mitochondrial NADH shuttling homologs, based on its cytosolic localization in all three media, *Pp*MdhB may be the homolog of *Sc*Mdh2 by shuttling NADH generated in the cytosol from glucose and methanol metabolism, and NADH generated in peroxisomes from oleate metabolism, to the mitochondria and may play a role in gluconeogenesis and the glyoxylate cycle (Figure 7).

However, based on its peroxisomal localization in oleate, *Pp*MdhB may also be the homolog of *Sc*Mdh3 in playing a role in β-oxidation by re-oxidizing NADH generated from fatty acid β-oxidation back to NAD^+^ to maintain the intraperoxisomal redox balance (Figure 7).

The peroxisomal and cytosolic locations of *Pp*MdhB in oleate satisfies the metabolic requirement wherein NADH that is produced in the peroxisome can be shuttled out of the peroxisome for redox balance and energy production. In contrast, there is no physiological need for Pp*MdhB* to be peroxisomal in methanol, wherein NADH is primarily produced in the cytosol (Figure 7).

### 3.2. Targeting of PpMdhB to Peroxisomes

Although *Pp*MdhB has a dual localization to the cytosol and peroxisome depending on the media, no recognizable PTS1 or PTS2 was found in *Pp*MdhB, despite our observation that the peroxisomal trafficking of *Pp*MdhB requires a free N-terminal end and depends on the PTS2 receptor, Pex7. Similar to *P. pastoris*, *Y. lipolytica* also has two *MDH* genes, *YlMDH1* producing a mitochondrial isoform with a strong mitochondrial prediction based on MitoProt II and *Yl*MDH2 encoding a varying ratio of cytosolic and peroxisomal isoforms (ending in a PTS1), due to alternative splicing depending on the media [17]. However, most *P. pastoris* genes do not have introns, only a single cDNA encoding *Pp*MdhB has been reported and there are no splice donor, acceptor, or branch point sites located near the 5′ or 3′ ends of the coding region to suggest that PpMdhB could use alternative splicing to acquire a PTS2. The small fraction of peroxisomal *Pp*MdhB observed in the oleate medium may be due to one of the multiple mechanisms alluded to earlier in the Discussion. We also cannot rule out the presence of a novel or noncanonical PTS in *Pp*MdhB that does not fit the known consensus for PTS1 or PTS2 sequences. There is precedent for noncanonical PTSs in several yeast proteins that use either the PTS1 receptor, Pex5 [30,31], or the PTS2 coreceptor, Pex20 [32], for entry into peroxisomes. Further work is necessary to determine which of the possible mechanisms is responsible for the peroxisomal targeting of *Pp*MdhB in oleate.

## 4. Materials and Methods

### 4.1. Yeast Strains and Media

Strains used are shown in Appendix A. Growth medium used in this study: YPD (2% glucose, 2% bacto-peptone, 1% yeast extract), YNB (0.17% yeast nitrogen base without amino acids and ammonium sulfate, 0.5% ammonium sulfate), CSM (complete synthetic medium of amino acids and supplements), glucose medium (2× YNB, 0.79g/L CSM, 0.04 mg/L biotin, 2% dextrose), oleate medium (2× YNB, 0.79g/L CSM, 0.04 mg/L biotin, 0.02% Tween-40, 0.2% oleate), and methanol medium (2× YNB, 0.79g/L CSM, 0.04 mg/L biotin, 0.5% methanol).

### 4.2. Plasmid Constructions

Plasmids, shown in Appendix A, were constructed by Gibson Assembly. Gibson assembly primers were designed using NEBuilder version 2.5.2 (https://nebuilder.neb.com/#!/ accessed on 14 March 2021). DNA was amplified from wild-type genomic DNA by PCR using Advantage 2 Polymerase (Cat#639201; Takara Bio USA, Mountain View, CA, USA). Plasmid backbones were double-digested with the necessary restriction enzymes and purified using the Qiagen DNA purification kit (QIAGEN GmbH, Hilden, Germany). DNA inserts were cloned into the digested vectors using the NEBuilder Hifi DNA Assembly Mastermix (E2621S; New England Biolabs, Ipswich, MA, USA) and incubated for 1 h at 50 °C. Newly-assembled plasmids were transformed into GC10 *E. coli* cells made chemically-competent. Then, 2 μL of the Gibson Assembly mixture was added to the 50 μL GC10 competent cells, placed on ice for 30 min, heat shocked for 30 s at 42 °C, placed on ice for 2 min, and then 950 μL of S.O.C. medium was added. The mixture was incubated at 37 °C for 1 h shaking at 250 rpm and then plated on LB plates with the appropriate antibiotic. Colonies were screened by colony PCR, and positive colonies were cultured in 5 mL of LB + antibiotic overnight. Plasmids were purified using the Wizard Miniprep Kit (Promega Corporation, Madison, WI, USA) and sequenced by Eton Bioscience.

### 4.3. Yeast Strain Constructions

Plasmids were linearized with the appropriate restriction enzyme before transforming into yeast cells made competent by electroporation. Yeast strains were cultured in a 250 mL flask with 50 mL of YPD shaking at 250 rpm at 30 °C and grown until 1–2 OD_600_/_mL_ (exponential phase). Cells were pelleted by centrifugation at 3000 rpm for 3 min and resuspended in 5 mL of YPD, 100 μL of 1M HEPES (pH 8), and 125 μL of 1M DTT. The resuspended cells were incubated at 30 °C for 15 min on a lab rotator and then washed three times with cold, sterile water and once with 1M cold sorbitol. Cells were gently resuspended with 200–300 μL of cold sorbitol and placed on ice for 1 h. Then, 50 μL of cells and 5 μL of linearized DNA were mixed together and transferred into a prechilled electroporation cuvette. After 10 min on ice, the mixture was electroporated using the BTX Harvard Apparatus Electroporator (BTX, San Diego, CA, USA) and was immediately resuspended in 1M cold sorbitol. The transformed cells were plated on YPD plates with the appropriate yeast selection markers and incubated at 30 °C for a few days. Colonies were screened by Western Blot or fluorescence microscopy.

### 4.4. Fluorescence Microscopy

Cells were grown in YPD at 30 °C until exponential phase (1–2 OD_600_/_mL_), washed twice with sterile water, and then transferred to peroxisome proliferation media or used for microscopy. Cells were grown in oleate or methanol media overnight (16h) and washed twice with sterile water. Cells were then pelleted, 1.5 μL of cells were mixed with 1% low melting point agarose and placed on a glass slide with a coverslip and imaged using 63× or 100× magnification on a Carl Zeiss Axioskop 2 MOT microscope (Carl Zeiss Microscopy, Gottingen, Germany). Images were taken on an AxioCam HRm digital camera(Carl Zeiss MicroImaging GmbH, Gottingen, Germany); no digital gain was used, exposition for mitochondria or peroxisome markers was adjusted as needed, exposition for BiFC ranged between 200 to 1000 ms and was kept constant during microscopy in different media. Images were processed using AxioVision software V4.8.2.0 (Carl Zeiss Microscopy, White Plains, NY, USA). The images are representative results from experiments conducted at least in triplicate. Methodology to determine peroxisome or mitochondrial localization in different background strains is as follows: Cells showing distinct peroxisome labeling with Pex3-mRFP or BFP-SKL or mitochondrial labeling with Tom20-mRFP were first marked in the red or blue channel in the AxioVision software. Then, these marks were analyzed for colocalization in the green channel with the GFP-tagged proteins or fluorescence obtained from BiFC. No quantification was assessed or deemed necessary as we observed either total or no colocalization.

## Figures and Tables

**Figure 1 ijms-22-04890-f001:**
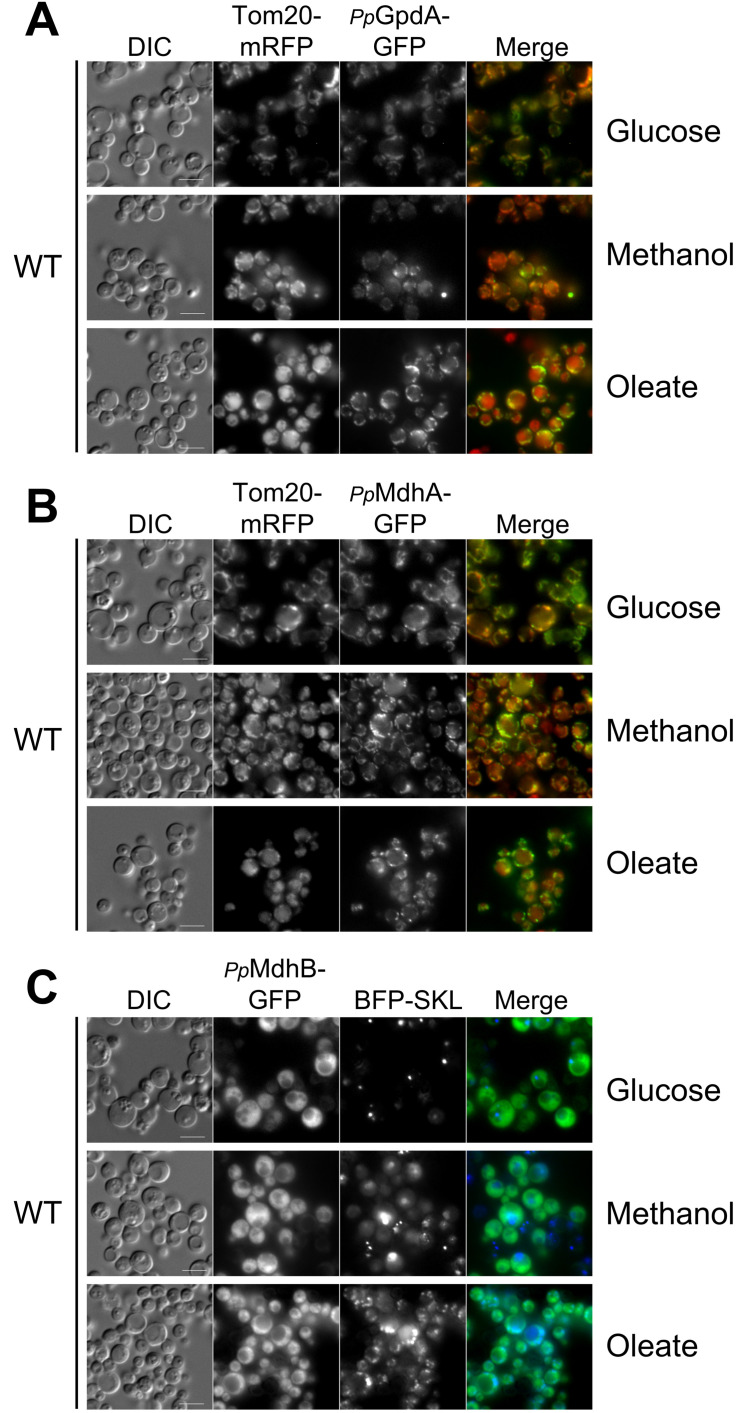
Localization of *P. pastoris* NADH-shuttling proteins fused with GFP. Fluorescence microscopy of strains grown in glucose, methanol, and oleate overnight. GFP was fused to the C-terminus of the NADH shuttling proteins, and the fusion proteins were expressed from their endogenous promoters. (**A**) Wild-type cells expressing PpGpdA-GFP, (**B**) Wild-type cells expressing PpMdhA-GFP, and (**C**) Wild-type cells expressing PpMdhB-GFP. Colocalization with the mitochondria and peroxisomes was observed with Tom20-mRFP or BFP-SKL as mitochondrial and peroxisomal markers, respectively. Bars: 5 μm.

**Figure 2 ijms-22-04890-f002:**
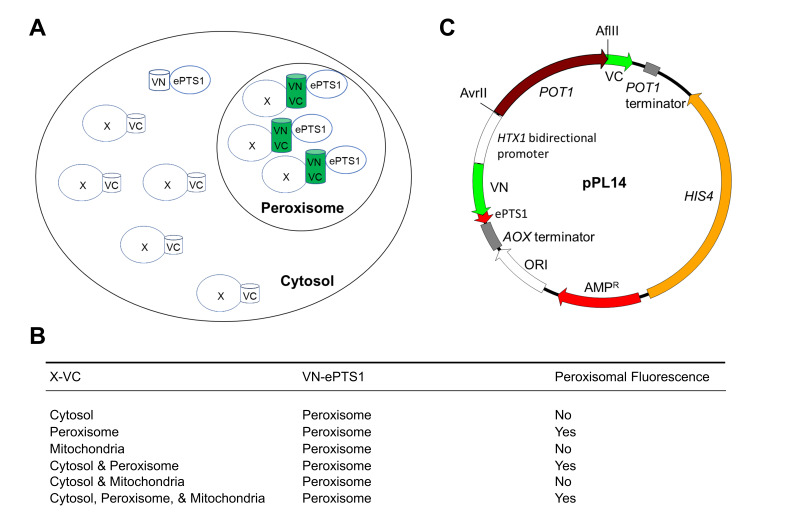
Expected fluorescence using the divergent BiFC assay. (**A**) Schematic representation of the expected outcome of the assay. (**B**) Potential subcellular localizations that may yield peroxisomal fluorescence. (**C**) Plasmid construct (VN and VC—BiFC moieties, *POT1* and *AOX* terminator—transcription terminator, *HIS4—*histidine selection marker for yeast, Amp^R^—ampicillin resistance gene, *HTX1* bidirectional promoter—strong constitutive bidirectional promoter, ePTS1—enhanced peroxisomal-targeting signal 1, *POT1*—thiolase open reading frame, ORI—bacterial replication origin). AvrII and AflII were used for double digestion to replace *POT1* with the gene encoding the protein of interest.

**Figure 3 ijms-22-04890-f003:**
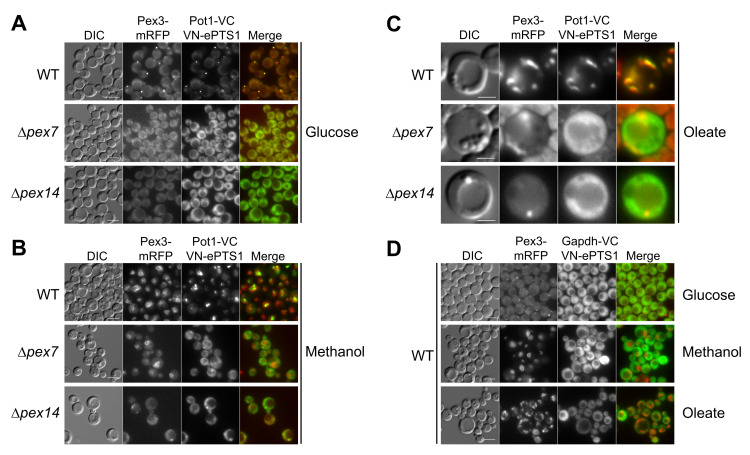
Feasibility of the divergent BiFC assay using known peroxisomal matrix and cytosolic proteins. The PTS2-containing protein, thiolase (Pot1), was fused to VC (Pot1-VC), and the PTS1 was fused to VN (VN-ePTS1). Strains were grown overnight in either (**A**) glucose, (**B**) methanol, or (**C**) oleate. (**D**) Strains were grown overnight in glucose, methanol, and oleate, and peroxisomal fluorescence were not observed for a cytosolic protein (Gapdh). Peroxisomes were visualized using Pex3-mRFP. Bars for (**A**,**B**,**D**) 5 μm; bars for (**C**) 2 μm.

**Figure 4 ijms-22-04890-f004:**
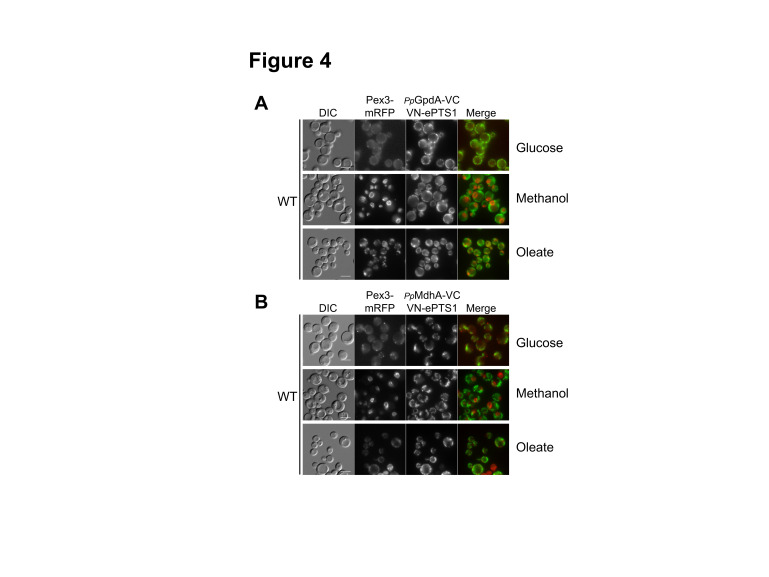
Localization of *Pp*GpdA and *Pp*MdhA in wild-type cells using the divergent BiFC assay. Strains were grown in glucose, methanol, and oleate overnight. Peroxisomes were visualized using Pex3-mRFP. (**A**) *Pp*GpdA-VC; (**B**) *Pp*MdhA-VC. Bars: 5 μm.

**Figure 5 ijms-22-04890-f005:**
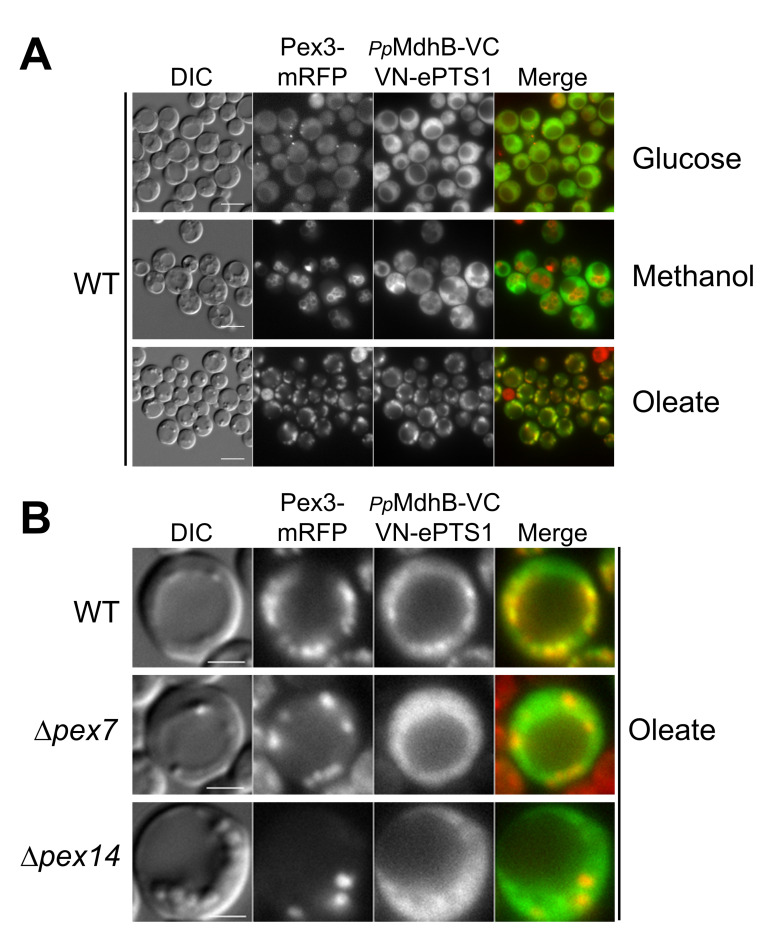
Localization of *Pp*MdhB using the divergent BiFC assay. (**A**) Wild-type cells were grown overnight in glucose, methanol, and oleate. Bars: 5 μm. (**B**) Wild-type, Δ*pex7,* and Δ*pex14* cells grown overnight in oleate. Bars: 2 μm. Peroxisomes were visualized using Pex3-mRFP.

**Figure 6 ijms-22-04890-f006:**
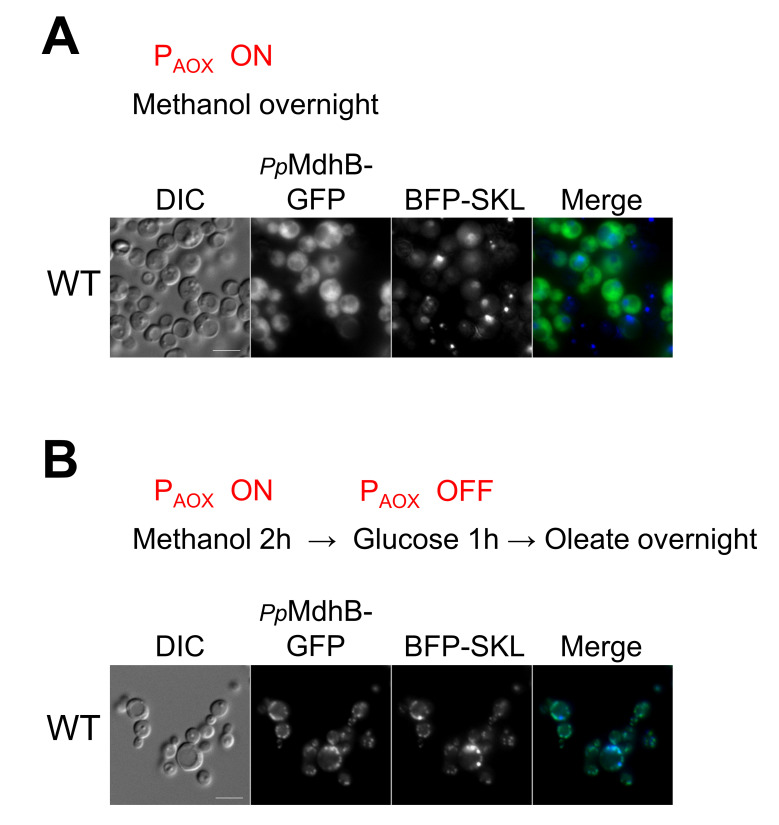
MdhB-GFP pulse-chase uncovers its localization to a small peroxisomal fraction. Fluorescence microscopy of *Pp*MdhB-GFP expressed from P_AOX_. (**A**) Strains were grown overnight in methanol. (**B**) Strains were grown for 2 h in methanol to express *Pp*MdhB-GFP (P_AOX_ ON), transferred to glucose for 1 h to repress the *AOX* promoter (P_AOX_ OFF), and grown overnight in oleate to induce peroxisome proliferation. Colocalization studies with the peroxisome were observed with BFP-SKL as peroxisomal markers, respectively. Bars: 5 μm.

**Figure 7 ijms-22-04890-f007:**
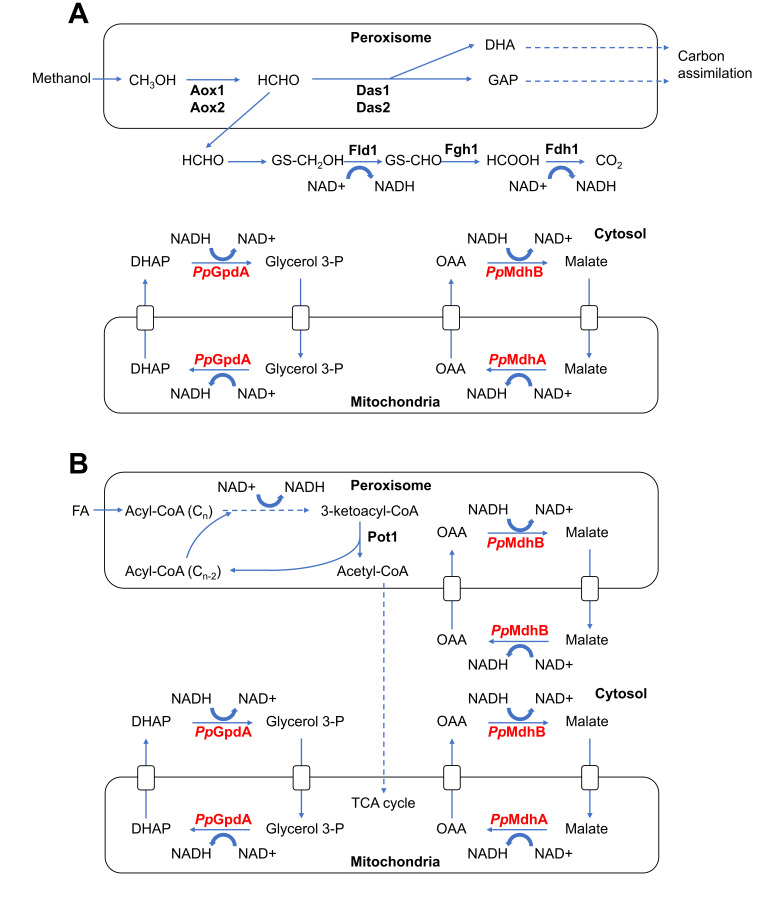
Proposed NADH shuttling in *P. pastoris*. (**A**) NADH shuttling in methanol. During methanol metabolism, methanol (CH_3_OH) is broken down into formaldehyde (HCHO), producing hydrogen peroxide (H_2_O_2_) as a byproduct. Some formaldehyde is converted in peroxisomes into dihydroxyacetone (DHA), and glyceraldehyde-3-phosphate (GAP) by dihydroxyacetone synthase 1 and 2 (Das1 and Das2), for carbon assimilation. Some formaldehyde diffuses into the cytosol, producing NADH through reactions involving Fld1 (formaldehyde dehydrogenase) and Fdh1 (formate dehydrogenase). NADH produced in the cytosol from methanol metabolism may be shuttled to the mitochondria for energy production through the malate or glycerol 3-phosphate (glycerol 3-P) shuttles. MdhB can convert oxaloacetate into malate, oxidizing NADH into NAD^+^. Malate enters the mitochondria through a transporter. MdhA converts malate back to oxaloacetate, reducing NAD^+^ to NADH. GpdA can convert dihydroxyacetone phosphate (DHAP) into Glycerol 3-P, oxidizing NADH into NAD^+^. Glycerol 3-P can shuttle into the mitochondria where GpdA can regenerate NADH. (**B**) NADH shuttling in oleate. Fatty acid (FA), is actively transported into peroxisomes and acetylated to acyl-CoA in an ATP-dependent reaction. The newly acylated fatty acid is then desaturated, consuming FAD^+^ and producing H_2_O_2_ as a byproduct. The newly formed, desaturated FA-CoA ester is hydrated, and then oxidized to form 3-ketoacyl-CoA and NADH. Finally, the 3-keto-acyl-CoA is cleaved by Pot1 (peroxisomal 3-oxyacyl-thiolase), releasing a molecule of acetyl-CoA and producing a fatty acyl-CoA, which is two carbons shorter than the substrate that entered the cycle. From this point, the product can restart the cycle. NADH produced in the peroxisomes during FA metabolism is shuttled out of the peroxisomes through the malate shuttle. MdhB converts oxalo-acetate into malate, regenerating NAD^+^. MdhB regenerates NADH in the cytosol and also shuttles NADH from the cytosol to the mitochondria. MdhB re-oxidizes NADH back to NAD^+^, reducing oxaloacetate to malate, which is transported into the mitochondria, where MdhA regenerates NADH in the mitochondria, converting malate back to oxaloacetate. The glycerol 3-P shuttle can also shuttle NADH from the cytosol to the mitochondria.

**Table 1 ijms-22-04890-t001:** In silico analysis of malate and glycerol 3-phosphate dehydrogenases in *P. pastoris* and *S. cerevisiae*. PTS1 scores were based on the C-terminal sequences of the dehydrogenases. Scores greater than 0 typically have a predictable PTS1. MitoProt scores were based on the N-terminal predictions for mitochondrial presequences. MitoProtII scores with a probability greater than 0.5 are more likely to be mitochondrial proteins.

Protein	C-Terminal	PTS1 Score	N-Terminal	MitoProt Score
*Sc*Mdh1	NIEKGVNFVASK	−48.5	MLSRVAKRAFSSTVANP	0.9910
*Sc*Mdh2	GLEFVASRSASS	−38.2	Not predictable	0.0376
*Sc*Mdh3	KGKSFILDSSKL	4.9	Not predictable	0.1859
*Pp*MdhA	IAKGQEFVKQNP	−37.7	MLSTIAKRQFSSSASTA	0.9549
*Pp*MdhB	NIAKGTAFIAGN	−52.6	Not predictable	0.1410
*Sc*Gpd1	PDMIEELDLHED	−101.6	MSAAADRLNLTSGHLNAGRKRS	0.3132
*Sc*Gpd2	PEMIEELDIDDE	−86.8	MLAVRRLTRYTFLKRTH	0.9932
*Pp*GpdA	FNKTEDVKHWED	−58.1	MYLTSTVRALPVHFFRSRHCIRT	0.6471

## Data Availability

No data other than those presented are applicable.

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
