# Peer review of "BiFC Method Based on Intraorganellar Protein Crowding Detects Oleate-Dependent Peroxisomal Targeting of Pichia pastoris Malate Dehydrogenase"

_ijms, 2021, doi:10.3390/ijms22094890_

Round 1

Reviewer 1 Report

The authors present a BiFC methods that allows an improved analysis of peroxisomal localization. Using this method, a partial localization of Pichia pastoris malate dehydrogenase B to the peroxisome is found (when cells are grown on oleate).

This is a very nice paper, well conduced and well written, that deserves publication with very little changes. I have only a few comments:

The only major point is probably the control shown in figure 3D: Gapdh, when VC-tagged, does not colocalize to the peroxisome in the presence of VN-ePTS1. In my opinion this is a very important control because it shows that VN-ePTS1 does not recruit its binding partner to the peroxisome. But does it? 1) Can we exclude that there is not such colocalization? 2) Can VC and VN even interact in this constallation if they were in the same compartment? – I suggest co-expressing Gapdh-VC with a cytosolic VN as a positive control for this negative control. 3) What happens when VC is on the other side of Gapdh? (Related concerns in experiment in figure S2 – Can VC be functionally tagged on both termini? 4) All experiments in figures 4 and 5 (where the protein of interest is not in the peroxisome) are implicit controls for this question, so I am not worried, but I somehow miss a discussion why the interaction of VC an VN is irreversible on one side, but does/may not support piggy-backing on the other.

When N-terminally tagged MdhB (VC-PpMdhB) is probed with and VN-SKL, VC-PpMdhB is not found in the peroxisome. The conclusion/subheading, however, “3.5. No PTS1 sequence targets PpMdhB to peroxisomes” appears too strong.

Line 355: “Thus, we expected that some of the NADH-shuttling proteins in P. pastoris would be peroxisomally localized,” – not clear to me how this could be concluded from the preceding statement.

Line 387: “Our work suggests that PpGpdA may be the homolog of ScGpd2 based on its mitochondrial localization” – Homology in the sense of common ancestry can be safely concluded from a sufficiently high level of sequence similarity. In that context, it would be interesting if the authors could report the (pairwise) sequence identity/similarity.

Figure 7: Spell out all abbreviations (e.g. Das, DHA, the latter could be confused with another famous peroxisome metabolite)

Refs 6 and 30: page or page range missing

Reviewer 2 Report

The work by Farré et al. studies the localization of different malate dehidrogenase isoforms in Yeast by using different microscopy-based approach and molecular biology techniques. The work is interesting for the field in terms of basic mechanisms underlying cell metabolism.

My major concern is that there is no description of the number of experiments, technical or biological replicates, number of cells analyzed, etc. Authors should include all this information in the manuscript and quantify all the observations to sustain their statements/conclusions.

Other Major points:

Authors must quantify their observations (Figure 1, 3, 4, 5, 6; partitioning between organelles and cytosol, etc.).

The details on the microscope, objectives, beam lines, software, image acquisition and processing of the images are mostly inexistent; a confocal microscope must be desirable for this kind of study; even essential. Instead, a wide-field combined with deconvolution could be used.

Once this has been performed and the data obtained coincide and support the statements raised within the present version of the manuscript, the work may be accepted.

Round 2

Reviewer 2 Report

I agree with the authors that the images they provide in their report now allow better observation of the results.

I would therefore ask the authors to use the images with the highest magnification in the figures of the manuscript and to include the images showing a large number of cells as supplemental figures. This will improve information given to readers.

I do not have any supplementary comments.

This is a very exhaustive work. Well done!